# High Thermoelectric Performance of a Novel *γ*-PbSnX_2_ (X = S, Se, Te) Monolayer: Predicted Using First Principles

**DOI:** 10.3390/nano13091519

**Published:** 2023-04-29

**Authors:** Changhao Ding, Zhifu Duan, Nannan Luo, Jiang Zeng, Wei Ren, Liming Tang, Keqiu Chen

**Affiliations:** Department of Applied Physics, School of Physics and Electronics, Hunan University, Changsha 410082, China; dingchanghao@hnu.edu.cn (C.D.); dzf1197333134@hnu.edu.cn (Z.D.); luonn@hnu.edu.cn (N.L.); jiangzeng@hnu.edu.cn (J.Z.); lmtang@hnu.edu.cn (L.T.)

**Keywords:** thermoelectric properties, first principles, Boltzmann transport equation

## Abstract

Two-dimensional (2D) group IV metal chalcogenides are potential candidates for thermoelectric (TE) applications due to their unique structural properties. In this paper, we predicted a 2D monolayer group IV metal chalcogenide semiconductor γ-PbSn_2_ (X = S, Se, Te), and first-principles calculations and Boltzmann transport theory were used to study the thermoelectric performance. We found that γ-PbSnX_2_ had an ultra-high carrier mobility of up to 4.04 × 10^3^ cm^2^ V^−1^ s^−1^, which produced metal-like electrical conductivity. Moreover, γ-PbSn_2_ not only has a very high Seebeck coefficient, which leads to a high power factor, but also shows an intrinsically low lattice thermal conductivity of 6–8 W/mK at room temperature. The lower lattice thermal conductivity and high power factors resulted in excellent thermoelectric performance. The ZT values of γ-PbSnS_2_ and γ-PbSnSe_2_ were as high as 2.65 and 2.96 at 900 K, respectively. The result suggests that the γ-PbSnX_2_ monolayer is a better candidates for excellent thermoelectric performance.

## 1. Introduction

The world’s energy demand is increasing due to the development of science and technology. Thermoelectric modules can directly convert electricity into thermal energy for cooling and heating and can also harvest waste heat for electrical power. They have, thus, attracted significant attention [1,2]. The performance of thermoelectric materials is usually evaluated by the dimensionless figure of merit (ZT), which is defined as $ZT=S^{2}\sigma T/\left(\kappa_{e}+\kappa_{l}\right)$, where *S* is the Seebeck coefficient, σ is the electrical conductivity, *T* is the absolute temperature, and $\kappa_{e}$ and $\kappa_{l}$ are the electronic thermal conductivity and the lattice thermal conductivity, respectively [3,4]. The higher the ZT value, the better the efficiency μ of the thermoelectric conversion. The efficiency of thermoelectric devices can reach 25% in power generation and exceed traditional refrigeration when ZT > 3 [5,6]. The definition of ZT suggests that high-performance thermoelectric devices depend on the increase in the power factor $PF=S^{2}\sigma$ (W·m^−1^ K^−1^) and the decrease in thermal conductivity.

However, due to the Wiedemann–Franz law, there is a coupling relationship between electronic transport coefficients, which makes it a significant challenge to improve ZT and enhance thermoelectric performance [7]. In inorganic thermoelectric materials, the improvement of the ZT value is a slow process until the paradigm of “phonon glass, electronic crystal” was first proposed by Slack et al. [8]. This paradigm indicates that we need to maintain the electrical conductance at a high level to maximize the ZT value while trying to reduce the thermal conductance. Subsequently, in order to improve thermoelectric performance, many advanced methods have been proposed, which can be divided into two main categories, namely, phonon engineering and electronic engineering, for optimizing phonon and electronic transport properties, respectively [5,9,10,11,12,13,14]. Phonon engineering is focused on the modulation of lattice thermal conductivity by suppressing the mean free path of phonons. Electron engineering aims at the modulation of the power factor and Seebeck coefficient under the optimum carrier concentration. This includes strain engineering [15,16,17,18,19,20], doping defects [21,22,23], molecular junction [4,24,25,26], superlattices [27,28,29], and heterostructures [30,31].

Two-dimensional materials with layered structures have attracted extensive attention as efficient thermoelectric materials due to their outstanding electronic and mechanical properties [32,33]. In addition, 2D materials exhibit great potential as thermoelectric candidates due to the enhanced Seebeck coefficient resulting from the increased density of states in proximity to the Fermi energy level [34,35]. In the past several decades, the thermoelectric performance of a series of 2D materials has been theoretically predicted and samples have been experimentally fabricated [36,37,38]. In particular, many group VI compounds, such as single layers of transition metal dichalcogenides (TMDCs), MX_2_ (M = Mo, W, Ti, and X = S, Se, Te), and group IV metal chalcogenides, AX_2_ (A = Ge, Sn, Pb, and X = S, Se, Te), have attracted attention in recent decades due to their unique semiconducting characteristics [35,39,40]. Recently, a novel class of 2D group IV metal chalcogenides, (AX)_2_ (A = Si, Ge, Sn, Pb; X = Se, Te), has been theoretically predicted through ab initio calculations [41]. Dong et al. reported that the γ-SnX (X = S, Se, or Te) has high thermoelectric performance due to the low thermal conductivity [14]. Then, Jia et al. reported high thermoelectric properties due to strong anharmonic effects in the (PbX)_2_ (X = S, Se, Te) monolayer [42]. Thus, the γ-phase group IV metal chalcogenides exhibit great potential as candidates for thermoelectric applications.

In this work, we predicted a series of 2D γ-phase group IV metal chalcogenides, namely, γ-PbSnX_2_ (X = S, Se, Te), based on the crystal structure of γ-AX (A = Pb, Sn and X = S, Se, Te). In addition, we investigated the thermoelectric transport properties using first-principles calculations combined with the Boltzmann transport equation. The density functional theory (DFT) has been widely used to predict the thermoelectric properties of materials [43,44]. The results show that these materials have high power factors and low lattice thermal conductivity, leading to a high figure of merit (ZT). The studies indicate that these materials are potential candidates for high-temperature thermoelectric materials.

## 2. Computational Method

In this work, we performed first-principles simulations using the Vienna Ab initio Simulation Package (VASP) based on density functional theory (DFT) [45,46] and using the projector augmented wave (PAW) pseudopotential [47,48,49,50]. The generalized gradient approximation (GGA) method with the Perdew–Burke–Ernzerhof (PBE) exchange–correlation (XC) functional was employed [51]. The total energy convergence criterion of $10^{-8}$ eV, the force convergence criterion of 0.001 eV/Å, and the kinetic energy cutoff of 500 eV were used to optimize the crystals [50]. A set of 15 × 15 × 1 Monkhorst-Pack k-points [52] was used to sample the Brillouin zone. A 20 Å vacuum layer was set in the direction of the *z*-axis to avoid the interaction of periodic layers along the *z*-axis and the DFT-D3 method was used to correct the van der Waals (VDW) interactions [53]. A hybrid functional (HSE06) was used to calculate the electronic band properties of materials. The thermoelectric transport coefficient was calculated by solving the Boltzmann transport equation with the BoltzTraP package [54]. The relaxation time was calculated using the deformation potential theory and effective mass approximation. A 3 × 3 × 1 supercell was used to calculate the second-order and third-order force constants of the materials through the finite displacement method. The cutoff radius of the third-order force constant was set as the sixth-nearest neighbors. Lattice thermal conductivity was obtained by solving the Boltzmann transport equation using ShengBTE, and a grid density of 60 × 60 × 1 k-points was used to ensure convergence [55].

## 3. Results

### 3.1. Structure and Stability

The monolayer γ-PbSnX_2_ (X = S, Se, Te) can be constructed from a γ-AX (A = Pb, Sn and X = S, Se, Te) monolayer by replacing one layer of chalcogen Pb/Sn atoms with another layer of chalcogen Sn/Pb atoms in the middle side, resulting in a hexagonal lattice structure with a P3m1 space group (Figure 1 and Appendix A). The calculated lattice constants of γ-PbSnS_2_, γ-PbSnSe_2_, and γ-PbSnTe_2_ are a = b = 3.96 Å, 4.11 Å, and 4.37 Å, respectively. The specific crystal structure parameters are shown in Table 1.

We next verified the structural stability of the γ-PbSnX_2_ (X = S, Se, Te) monolayer, including the mechanical, dynamic, and thermal stabilities. For the mechanical stability, we used the Born criterion of 2D materials as C_11_ > |C_12_| > 0 and C_66_ > 0 [56]. The calculated elastic constants of γ-PbSnX_2_, γ-PbSnX_2_, and γ-PbSnX_2_ were C_11_ = 39.5, 42.7 and 43.5 N/m, C_12_ = 12.9, 14.5, and 12.7 N/m, and C_66_ = 13.2, 14.1, and 15.4 N/m, respectively (Appendix A). These calculated values are given in Table 1 and satisfy the Born criterion. For the dynamic stability, we calculated the phonon dispersion of γ-PbSnX_2_. Figure 2a–c shows the phonon dispersion curve of γ-PbSnX_2_. Each unit cell of the γ-PbSnX_2_ monolayer has four atoms, with three acoustic and nine optical branches. The phonon frequencies of the γ-PbSnX_2_ monolayer are all positive, indicating the dynamic stability of the γ-PbSnX_2_ monolayer. They all have very low phonon frequencies and lead to a decrease in their phonon frequencies as the atomic mass of sulfur group elements increases. More interestingly, an apparent coupling occurs between optical and acoustic phonon modes in γ-PbSnX_2_ monolayers, which might lead to a low lattice thermal conductivity because of anharmonic scattering. To determine the thermal stability, we used ab initio molecular dynamics (AIMD) simulations to determine the stability of γ-PbSnX_2_ at 900 K. A 4 × 4 × 1 supercell was used for the AIMD simulation. The AIMD simulation results are shown in Figure 2d–f. The total energy was almost unchanged at a temperature of 900 K for 10 ps. In addition, the structure exhibited no obvious deformation at 900 K. The results confirm the thermal stability of γ-PbSnX_2_ at 900 K. In additional, we also calculated the formation enthalpy (ΔH) through:(1)$$\Delta H=\frac{H_{PbSnX_{2}}-H_{Pb}-H_{Sn}-2H_{X}}{4}$$
where $H_{PbSnX_{2}}$ is the total energy of the γ-PbSnX_2_ monolayer, $H_{Pb}$, $H_{Sn}$, and $H_{X}$ are the energy of a single atom (Pb, Sn, and X = S, Se, Te) of the structure.

According to Equation (Equation 1), it is obvious that negative formation energies are related exothermic chemical reactions, which imply stable products. As listed in Appendix A), the formation enthalpy values of γ-PbSnS_2_, γ-PbSnSe_2_, and γ-PbSnTe_2_ are −0.45, −0.37, and −0.23 eV/atom, respectively, which indicates that all the monolayers are stable. In addition, it can be seen from Appendix A that the lighter structures of atoms are more stable because they are more likely to form through exothermic reactions.

In conclusion, we determined the stability of materials by calculating their formation enthalpy, mechanical stability, dynamic stability, and thermal stability. The formation enthalpy of the material calculated using Equation (Equation 1) is negative, which indicates that the material can release energy during chemical formation and reach a stable state. The phonon dispersion curve can reflect the dynamic stability of the structure, indicating that the material is dynamically stable. The AIMD simulation results can reflect the thermal stability of the structure at 900K, indicating that the structure can exist stably at 900 K. These results indicate that these structures are stable.

### 3.2. Electronic Band Structure

Figure 3 shows the electronic band structures of the γ-PbSnX_2_ (X = S, Se, Te) monolayers calculated using the PBE and HSE06 exchange–correlation functionals. The corresponding band gap values are given in Table 2. They are both indirect band gaps with a conduction band minimum (CBM) at the high symmetry point Γ (0, 0, 0) and a valence band maximum (VBM) between the high symmetry points Γ (0,0,0) and K (1/3, 1/3, 0). The band gap calculated using HSE06 is larger than that calculated using PBE. It is noted that the PBE functional often underestimated the band gap value, while the HSE functional gave a reliable band gap value compared with the experiment. The band gaps of γ-PbSnS_2_, γ-PbSnSe_2_, and γ-PbSnTe_2_ were 0.86 (1.37) eV, 0.63 (1.08) eV and 0.61 (0.98) eV, respectively. The band gaps calculated by all the methods gradually decreased as the atomic number of the substituted chalcogenide element (S, Se, and Te) increased. The thermoelectric characteristics of the γ-PbSnX_2_ monolayer can be easily tuned at a suitable doping concentration for 2D materials, according to such a moderate band gap. The band types and shapes calculated using the PBE and HSE functionals were essentially unchanged except for the band gap. The variable situation theory and effective mass approximation method were used to calculate the carrier mobility and relaxation time of the material, and the band values near the VBM and CBM were used for fitting. However, the band structure and shape calculated using the PBE and HSE06 functionals in the VBM and CBM were the same; only the band gap was different. This means that the dispersion relationship between the PBE and HSE functionals was consistent and the effective mass calculated using the PBE and HSE06 functionals was assumed to be the same. Therefore, we used the PBE function to obtain the effective mass and carrier mobility of these materials. The partial density of states (PDOS) of the γ-PbSnX${}_{2}$ is shown in Figure 3d–f, with the valence bands closer to the Fermi energy level. The valence bands around the Fermi level originate from the S, Se, and Te atoms, and the conduction bands are jointly contributed by Sn or Pb, S, Se, and Te atoms. Figure 3d–f show that γ-PbSnS${}_{2}$, γ-PbSnSe${}_{2}$, and γ-PbSnTe${}_{2}$ all have a very sharp density of state peaks at the Fermi energy level attachment, where γ-PbSnS${}_{2}$ has a higher density of state peaks than γ-PbSnSe${}_{2}$ and γ-PbSnTe${}_{2}$.

### 3.3. Carrier Mobility and Relaxation Time

We estimated the electrical characteristics of the γ-PbSnX${}_{2}$ (X = S, Se, Te) monolayer using the BoltzTraP package, which is based on the semi-classical Boltzmann transport equation. Additionally, the BoltzTraP2 package uses constant relaxation time approximation, which means the calculated results are divided by the relaxation time. Therefore, in order to accurately calculate the thermoelectric properties of the material, the relaxation time of the material must be obtained.

Here, the carrier mobilities and relaxation time of the 2D materials are calculated using deformation potential theory and the effective mass approximation method [57,58,59]: (2)$$\mu_{2D}=\frac{e\hbar^{3}C_{2D}}{k_{B}T|m^{*}{|}^{2}{E_{1}}^{2}}$$
(3)$$\tau =\frac{\hbar^{3}C_{2D}}{k_{B}T\left|m^{*}\right|{E_{1}}^{2}}$$
where *ℏ* is the reduced Planck constant, k${}_{B}$ is the Boltzmann constant, T is the temperature, $\mu_{2D}$ is the carrier mobility, and τ is the relaxation time. C${}_{2D}$ is the elastic constant defined by $C_{2D}=\frac{\partial^{2}E_{total}/\partial \varepsilon^{2}}{S_{0}}$, where E${}_{total}$ is the total energy after applying a uniaxial term ($\varepsilon =\Delta l/l_{0}$), and S${}_{0}$ is the area at equilibrium. $m^{*}$ is the effective mass defined by $m^{*}=\hbar^{2}{\left(\frac{d^{2}\varepsilon \left(k\right)}{dk^{2}}\right)}^{-1}$; we calculated the effective mass using vaspkit code [60]. Six discrete points were selected in the CBM and VBM attachments for polynomial fitting, and the truncation radius was 0.015/Å to 0.01/Å. With the decrease in the truncation radius, Δk also decreased, but the results of the effective mass calculation were almost constant. Therefore, we believe that the results converged, that is, smaller Δk values would not have changed the reported effective mass. $E_{1}$ is the deformation potential constants defined by $E_{d}=\Delta E/\Delta \varepsilon$, where ΔE is the energy shift of the band edge of CBM or VBM with respect to the vacuum level (Appendix A).

Table 3 shows the results of the electric and hole carrier mobilities calculated with the theory of deformation potential at 300 K. Among them, γ-PbSnS${}_{2}$ and γ-PbSnSe${}_{2}$ had a ultra-high hole carrier mobility; in particular, γ-PbSnS${}_{2}$ had the highest hole mobility of 4.04 × 10${}^{3}$ cm${}^{2}$ V${}^{-1}$ s${}^{-1}$, which was significantly higher than that of other two-dimensional semiconductors, e.g., MoS${}_{2}$ (285 cm${}^{2}$ V${}^{-1}$ s${}^{-1}$) [61], SnS${}_{2}$ (756 cm${}^{2}$ V${}^{-1}$ s${}^{-1}$) [62], SnSe${}_{2}$ (462 cm${}^{2}$ V${}^{-1}$ s${}^{-1}$) [62], γ-PbX_2_ (780 cm^2^ V^−1^ s^−1^) [42], and γ-SnX_2_ (1364 cm^2^ V^−1^ s^−1^) [14]. The high carrier mobility of γ-PbSnS_2_ and γ-PbSnSe_2_ was due to a combination of a low effective mass and deformation potential energy. The high hole carrier mobility indicates that γ-PbSnS_2_ and γ-PbSnSe_2_ are potential p-type semiconductors. In addition, Hung et al. showed that a high carrier mobility is one of the important parameters for screening thermoelectric materials [63,64]. The ultra-high carrier mobility indicates that γ-PbSnX_2_ has excellent hole transport properties and is an excellent thermoelectric material.

### 3.4. Thermoelectric Properties

We investigated the thermoelectric properties of the materials in the temperature range of 300 K to 900 K. As shown in Figure 4a–c, all three materials had a high Seebeck coefficient at 300 K. Among them, the Seebeck coefficients of γ-PbSnS_2_, γ-PbSnSe_2_, and γ-PbSnTe_2_ were 1400 μV/K, 800 μV/K, and 900 μV/K, respectively, which were much higher than those of other common 2D materials, such as SnTe (600 μV/K) [65], MoSe_2_ (427 μV/K) [66], and WS_2_ (328 μV/K) [67]. The Seebeck coefficient of γ-PbSnS_2_ was higher than those of γ-PbSnSe_2_ and γ-PbSnTe_2_. This is because γ-PbSnS_2_ has a higher density of state peaks than γ-PbSnSe_2_ and γ-PbSnTe_2_ near the Fermi energy level (as shown in Figure 3), and the Seebeck coefficient is proportional to the density of state peaks: S∝d(DOS)/dE. The high density of state peaks near the Fermi energy level indicates that the material will have a higher Seebeck coefficient. The high Seebeck coefficient indicates that these materials may have high thermoelectric properties.

Figure 4d–f show the electrical conductivities σ obtained by multiplying σ/τ by τ, where σ/τ was calculated using the BoltzTraP package and τ is the relaxation time calculated using deformation potential theory. The electrical conductivity of γ-PbSnX${}_{2}$ (X = S, Se, Te) exhibited a similar behavior at different temperatures and decreased as the temperature increased. This was caused by the enhanced lattice vibrations and electron scattering at high temperature. The electric conductivity of γ-PbSnX${}_{2}$ gradually decreased as the atomic number of the substituted chalcogenide element (S, Se, and Te) increased. In addition, the value of the conductivity is related to the carrier concentration. The conductivity of γ-PbSnX${}_{2}$ was higher in the positive carrier concentration range, which shows the characteristics of p-type semiconductors. It is worth noting that the electrical conductivity of γ-PbSnX${}_{2}$ reached the same order (10${}^{6}$–10${}^{7}$/Ωm) as that of metals due to its ultra-high carrier mobility. The high Seebeck coefficient and conductivity indicate that they have a high power factor ($PF=S^{2}\sigma$), which is important for thermoelectric devices. With the Seebeck coefficient and electrical conductivity, we evaluated the power factor (PF) of γ-PbSnX_2_ (X = S, Se, Te), as shown in Figure 4g–i. All three materials produced the same trend for the power factor, which decreased with the increase in temperature. The γ-PbSnS_2_ had the highest PF because of its higher Seebeck coefficient and electrical conductivity. Moreover, the PF values of these materials were maximum in the negative range due to their p-type properties. The PF is one of the important factors in evaluating the performance of thermoelectric materials, so a high PF predicts that the material will have high thermoelectric properties.

Materials with high thermoelectric properties require low thermal conductivity in addition to a high power factor. Figure 5 shows the phonon transport properties of γ-PbSnX_2_ (X = S, Se, Te). The results show that both γ-PbSnX_2_ have a very low thermal conductivity, which is about 6–8 W/mK at room temperature, and the value of the lattice thermal conductivity decreased with increasing temperature. In order to understand the lattice thermal conductivity of the γ-PbSnX_2_ monolayer, we explored the phonon-related properties, such as the phonon group velocity and anharmonic scattering rates. As shown in Figure 5a–c, the three acoustic phonon branches (ZA/TA/LA) become entangled and strong coupling occurs between the optical and acoustic phonon modes, which can strengthen the phonon scattering mechanism and, thus, lower $\kappa_{l}$. Figure 5d–f show the phonon group velocities of γ-PbSnX_2_, and it can be seen that they are both low, and low group velocities can lead to a low lattice thermal conductivity. The inverse of the phonon relaxation time in the relaxation time approximation (RTA) is equal to the total scattering rate, which is the sum of the isotopic scattering rate ($\tau_{i}^{1}$), the boundary scattering rate ($\tau_{b}^{1}$), and the anharmonic scattering rate ($\tau_{a}^{1}$). In general, a higher value indicates stronger phonon–phonon scattering and a lower phonon relaxation time, which is beneficial for reducing the lattice thermal conductivity. As shown in Figure 5g–i, the γ-PbSnX_2_ monolayer exhibited high phonon–phonon scattering rates (anharmonic scattering rates) and they are relatively close to each other. In addition, the value of the lattice thermal conductivity increased with the atomic number of the elements (S, Se, and Te), which is mainly due to the increase in the atomic mass and the decrease in the phonon frequency. In conclusion, the strong coupling between phonons, lower phonon group velocities, and higher scattering rates lead to lower lattice thermal conductivity, indicating that γ-PbSnX_2_ may be a suitable thermoelectric material.

After all transport coefficients were obtained, the dependence of ZT on the carrier concentration of the γ-PbSnX_2_ monolayer at different temperatures was calculated, as shown in Figure 6. All γ-PbSnX_2_ monolayers exhibited the maximum thermoelectric properties in the positive carrier concentration range due to their higher hole mobility. The ZT values for all three materials showed the same trend: increasing with increasing temperature. Among them, the ZT value of γ-PbSnTe_2_ was lower, reaching a maximum of only 1.4 at 900 K, which is due to its low power factor. In addition, γ-PbSnS_2_ and γ-PbSnSe_2_ reached ultra-high ZT values of 2.65 and 2.96 at 900 K due to a combination of their low lattice thermal conductivity and high power factor. These ultra-high ZT values for γ-PbSnX_2_ contribute to their low thermal conductivity along with their high power factor. High ZT values indicate that γ-PbSnS_2_ and γ-PbSnSe_2_ are high temperature thermoelectric materials with a good performance.

## 4. Conclusions

In this study, we constructed a series of γ-PbSnX_2_ (X = S, Se, Te) 2D monolayer material based on γ-(AX)_2_ (A = Sn, Pb and X = S, Se, Te) and calculated their thermoelectric properties using Boltzmann transport theory combined with first principles. The results show that these materials are narrow bandgap semiconductors with bandgap values of 0.98–1.37 eV according to the HSE06 functional. They have high hole carrier mobilities; in particular, γ-PbSnS_2_ has a hole carrier mobility of 4.04 × 10^3^ cm^2^ V^−1^ s^−1^. In addition, they all have high Seebeck coefficients and intrinsically low lattice thermal conductivities, which are 800–1400 μV/K and 6–8 W/mK at temperatures of 300 K. The ZT values of γ-PbSnS_2_, γ-PbSnSe_2_, and γ-PbSnTe_2_ at 900K were 2.65, 2.96, and 1.36, respectively. This high thermoelectric performance indicates that γ-PbSnX_2_ monolayer materials are excellent thermoelectric materials. Our theoretical study may help to optimize the thermoelectric properties of such materials.

## Figures and Tables

**Figure 1 nanomaterials-13-01519-f001:**
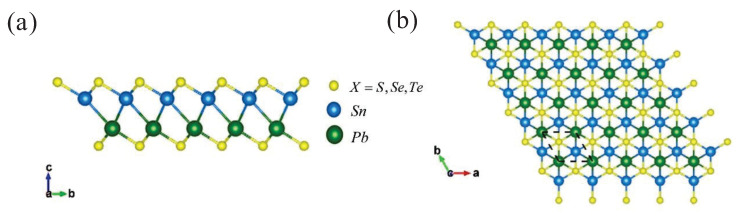
(**a**) Side view and (**b**) top view of the optimized γ-PbSnX_2_ (X = S, Se, Te) monolayer structure.

**Figure 2 nanomaterials-13-01519-f002:**
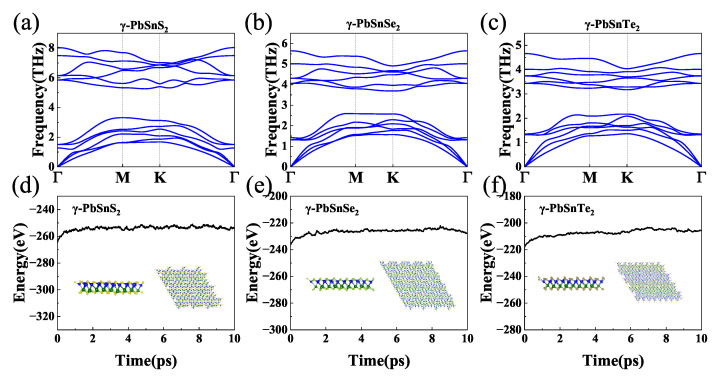
(**a**–**c**) Phonon dispersions for γ-PbSnX_2_ (X = S, Se, Te). (**d**–**f**) Total potential energy fluctuation and structures in AIMD simulations of γ-PbSnX_2_ (X = S, Se, Te).

**Figure 3 nanomaterials-13-01519-f003:**
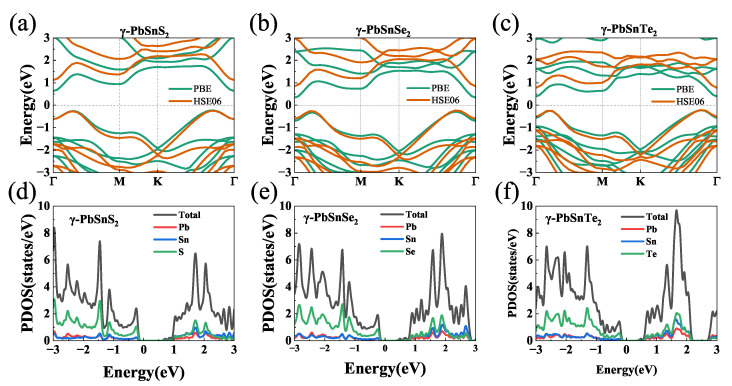
(**a**–**c**) Band structures of γ-PbSnX_2_ (X = S, Se, Te) calculated using the PBE and HSE06 functionals. (**d**–**f**) Partial density of states (PDOS) of the γ-PbSnX_2_. The Fermi level was set as zero.

**Figure 4 nanomaterials-13-01519-f004:**
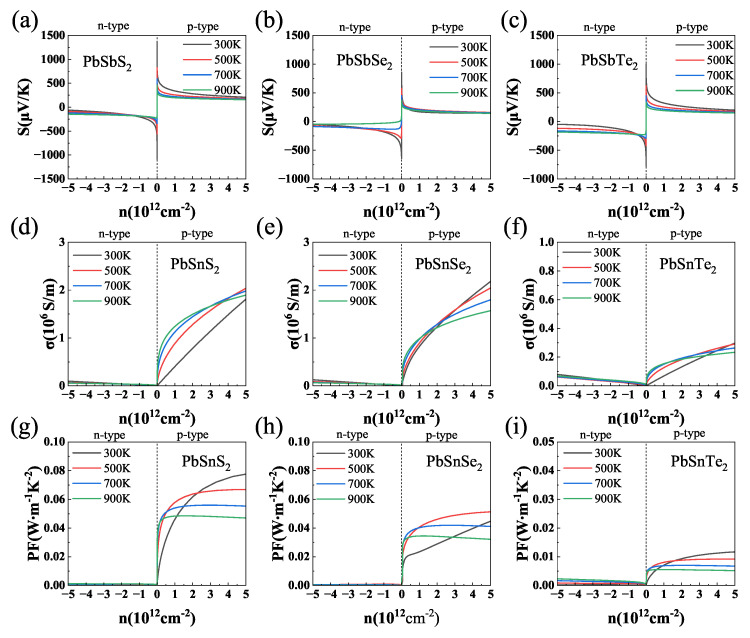
(**a**–**c**) Seebeck coefficients, (**d**–**f**) electrical conductivity, and (**g**–**i**) power factor (PF) of γ-PbSnX_2_ (X = S, Se, Te) as a function of carrier concentration.

**Figure 5 nanomaterials-13-01519-f005:**
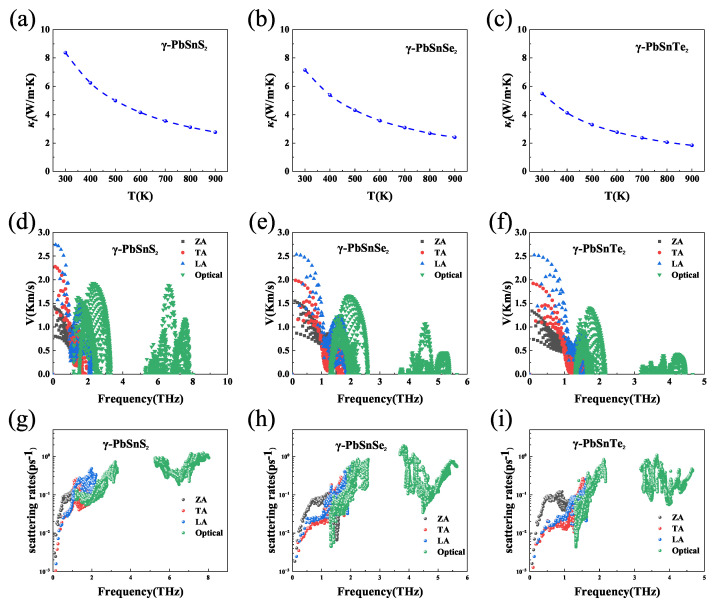
(**a**–**c**) Lattice thermal conductivity at different temperatures, (**d**–**f**) phonon group velocity at 300 K, and (**g**–**i**) anharmonic scattering rates at 300 K of γ-PbSnX_2_ (X = S, Se, Te).

**Figure 6 nanomaterials-13-01519-f006:**
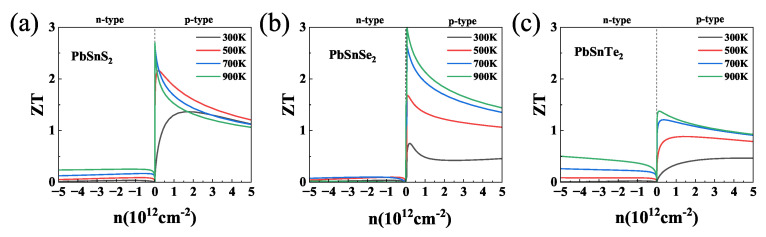
(**a**–**c**) The ZT values of γ-PbSnX_2_ (X = S, Se, Te) as a function of carrier concentration at 300 K.

**Table 1 nanomaterials-13-01519-t001:** Structural parameters for the γ-PbSnX_2_ (X = S, Se, Te). Here, *a* is the lattice constant, $d_{Sn-X}$, $d_{Pb-X}$, and $d_{Pb-Sn}$ are the Sn-X, Pb-X, and Pb-Sn bond lengths, respectively, and *h* is the vertical distance between the two outermost X atoms in Angstroms, as shown in Figure 1.

Material	*a* (Å)	$d_{\mathrm{Sn}-X}$ (Å)	$d_{\mathrm{Pb}-X}$ (Å)	$d_{\mathrm{Pb}-\mathrm{Sn}}$ (Å)	*h* (Å)	$C_{11}$ (N/m)	$C_{12}$ (N/m)	$C_{66}$ (N/m)
PbSnS_2_	3.96	2.64	2.68	3.58	5.49	39.5	12.9	13.2
PbSnSe_2_	4.11	2.77	2.81	3.51	5.52	42.7	14.5	14.1
PbSnTe_2_	4.37	2.97	3.01	3.46	5.59	43.5	12.7	15.4

**Table 2 nanomaterials-13-01519-t002:** Bandgap of γ-PbSnX_2_ (X = S, Se, Te) calculated using PBE and HSE06.

Material	Structure	Gap-Type	E_*g*_ (PBE)	E_*g*_ (HSE06)
PbSnS_2_	hexagonal (2D)	Indirect	0.86 eV	1.37 eV
PbSnSe_2_	hexagonal (2D)	Indirect	0.63 eV	1.08 eV
PbSnTe_2_	hexagonal (2D)	Indirect	0.61 eV	0.98 eV

**Table 3 nanomaterials-13-01519-t003:** Calculated effective mass ($m^{*}$), elastic constant ($C_{2D}$), deformation potential ($E_{1}$), carrier mobility ($\mu_{2D}$), and relaxation time (τ) for electrons (e) and holes (h) in the γ-PbSnX_2_ monolayer at 300 K.

Material	Carrier	$C_{2D}$ (N/m)	$m^{*}/m_{0}$	$E_{1}$(eV)	$\mu_{2D}$ (×10^3^ cm^2^ V^−1^ s^−1^)	τ (ps)
PbSnS_2_	e	39.5	0.22	4.92	0.476	0.059
	h		0.50	1.02	4.04	1.14
PbSnSe_2_	e	42.75	0.18	5.52	0.654	0.067
	h		0.357	2.66	1.421	0.288
PbSnTe_2_	e	43.55	0.23	6.52	0.256	0.033
	h		0.596	4.14	0.245	0.083

## Data Availability

Not applicable.

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
