# Peer review of "High Thermoelectric Performance of a Novel γ-PbSnX2 (X = S, Se, Te) Monolayer: Predicted Using First Principles"

_nanomaterials, 2023, doi:10.3390/nano13091519_

Round 1
Reviewer 1 Report
This paper reports a set of density functional calculations for three two-dimensional materials regarding all factors affecting their thermoelectric performance. The predicted thermoelectric figure of merit is rather good. However, before trusting the results in the manuscript, some necessary computational settings that were not disclosed in the manuscript should be added. Without detailed knowledge of these settings, a full assessment of the results is impossible.
1. The absence of negative phonon frequencies or structural integrity during high-temperature ab initio molecular dynamics, although promising indicators, do not establish structural stability. A comprehensive comparison with competing phases should be provided under all possible values for chemical potentials.
2. The switch between GGA and hybrid functionals has not been justified in the manuscript. If GGA was not accurate enough, why was it used at all? Even if it was less resource thirsty.
3. The procedure for calculating the effective masses should be further elaborated. Notably, the Δk used in differentiating the band structure around the extrema point should be stated and shown that the calculated effective masses converged with respect to this value. In other words, smaller Δk values would not have changed the reported effective masses.
4. The values in Figures 4 and 6, conductivities, Seebeck coefficient, and the figure of merit, should be reported as a function of carrier concentration instead of the shift in the Fermi level. Carrier concentrations are more easily understood by both theoreticians and experimentalists. Moreover, it is not readily understandable if the shifts shown can be achieved with experimentally reasonable carrier doping.
Other points:
5. The use of the word “constructed” in the abstract is misleading in the context of this paper. Since all results are theoretical, a more suitable word would be “predicted.”
6. The general suitability of the density functional methods in predicting thermoelectric performance was generally missing in the introduction:
Gorai et al. Computationally guided discovery of thermoelectric materials, Nature Reviews Materials 2, 17053 (2017); https://doi.org/10.1038/natrevmats.2017.53
Eivari et al. Low thermal conductivity: Fundamentals and theoretical aspects in thermoelectric applications, Materials Today Energy 21, 100744 (2021); https://doi.org/10.1016/j.mtener.2021.100744
A comerhensive edit is required.
Reviewer 2 Report
Authors conducted periodic DFT calculations of interesting 2D monolayer compounds, gama-PbSn2 (where X=S, Se, Te).
In their work, authors predicted (using VASP software) its structure parameters, thermoelectric properties, carrier mobility, electronic band structure.
This is promising theoretical study which could help experimentalists in their search for new materials with desired thermoelectrical properties.
Just minor issues:
i) please specify the type of employed pseudopotentials in your calculations;
ii) please provide supplementary material with obtained structures and other calculated properties.
Round 2
Reviewer 1 Report
Given the revisions, the paper can be accepted at this stage.
The paper is mostly intelligible and clear.